# Urban Violence, Migration and Alcohol, Tobacco, and Marijuana Use among Transnational Students in Northern Mexico

**DOI:** 10.3390/ijerph21010043

**Published:** 2023-12-27

**Authors:** Hilda García-Pérez, Stephen S. Kulis, Flavio F. Marsiglia, Paul A. Estabrooks

**Affiliations:** 1El Colegio de la Frontera Norte, Unidad Nogales, Reforma Av. No. 528 Col. del Rosario, Nogales 84020, Sonora, Mexico; 2School of Social and Family Dynamics, Arizona State University, P.O. Box 873701, Tempe, AZ 85287-3701, USA; kulis@asu.edu; 3Global Center for Applied Health Research (GCAHR), University Center 720, Arizona State University, Phoenix, AZ 85004-3920, USA; marsiglia@asu.edu; 4Department of Health & Kinesiology, University of Utah, 248 HPER North, Salt Lake City, UT 84112, USA; paul.estabrooks@health.utah.edu

**Keywords:** migrant adolescent, substance use, health-related risk behaviors, urban violence, prevention

## Abstract

This article reports on the findings of a study of the relationship between transnational experiences in the United States (US) and the use of alcohol, tobacco, and marijuana among 7th grade students (*n* = 1418). The study was guided by a cross-national framework for research on immigrant health and assessed the accumulation of risk factors for transnational adolescents. Data came from a survey conducted in 2017 in Nogales, Mexico. In this study, the last 30-day prevalence of use of alcohol, tobacco, and marijuana among students was 21.7%, 8.3%, and 2.4%, respectively. Most students were born in Nogales (69.6%), while 10.5% were born in the US, 7.5% attended school in the US, and 3.6% engaged in health-related risk behaviors while living in or visiting the US. Students with transnational experiences, such as attending school in the US, reported the highest 30-day prevalence of tobacco (13.3%) and marijuana (9.5%) use. After adjusting for family, school, access to substances and neighborhood violence variables, students who engaged in health-related risk behavior in the US had significantly increased odds of alcohol and marijuana use while later attending school in Mexico. The article discusses the findings from a prevention science perspective and provides implications for policy, practice, and future research on the Mexico-US border region.

## 1. Introduction

Transnational populations—groups who migrate and reside for appreciable periods in the receiving country while maintaining strong ties to their country of origin—are growing in size and prevalence. Since 2008, more than 2.8 million immigrants to the United States (US) have been deported to Mexico, including nearly a half million children. Most of these transnational children who are now enrolled in Mexican schools were born in the US. Many face diverse integration barriers in Mexican schools due to a lack of cultural identification, limited Spanish proficiency, and the educational system’s lack of attention to their challenges [1,2]. They are transnational students who do not fully belong to one country because their families left the US, voluntarily or involuntarily, and the youth are now enrolled in Mexican schools [3]. Once in Mexico, these students often experience negative educational outcomes, such as higher dropout rates and a greater likelihood of being behind a grade [1,4]. A study conducted in middle schools in a Mexican border context reported an association between length of schooling in the US and a dislike of Mexican schools among transnational students [1].

While deportation of parents puts their children at risk of economic instability and can negatively impact children’s long-term health and development [5], arriving in Mexican cities with high rates of community violence might also have a negative effect on the children’s mental health and well-being [6]. The confluence of mixed flows of migrants to northern Mexican border cities has led to the sudden arrival of transnational children and adolescents to an area where drug trafficking and violence converge [7]. These social contexts can amplify adolescents’ health risks during a critical developmental stage [8,9].

Most northern Mexican border states have been ranked among the 10 least peaceful places in the country [10]. At the same time, northern border communities have been the focus of international attention for the level of violence and drug trafficking-related activities [11,12]. Violence and substance use are serious health concerns for the populations of border communities since the area is a strategic route for trafficking of illegal drugs to the US [9,13,14,15].

### The Risk-Accumulation Model

We used the cross-national framework for research on immigrant health to assess the accumulation of risk factors [16]. This framework proposes that migrants’ health outcomes are influenced by risks accumulated in the place of origin, during their displacement, and in the place of arrival. This interplay between exposure to risk in the sending and receiving countries is critical to understanding the health of transnational children who are also exposed to a variety of risk factors at an early age during the different migration stages they go through as members of migrant households [16].

Although several studies have shown an association between substance use across different international migration stages in adults, data among adolescents are scarce. The available information suggests complex substance use patterns—i.e., initiation, current use and cessation—across migration stages [17,18,19]. For instance, a study conducted in Mexico found that having worked in the US or having a family member currently working in that country was associated with an increase in the likelihood of lifetime use of alcohol, marijuana, or cocaine [20]. Likewise, Zhang et al. [19] found that Mexican migrants, particularly those who were undocumented or male, were more likely to use illicit drugs, both in the destination country (US) and when they returned to Mexico. Similarly, Borges et al. [18] reported that Mexican migrants in the US were more likely to initiate the use of illicit drugs, but not alcohol, than non-migrants who remained in Mexico.

On the other hand, a study that explored the relationship between migration intentions and alcohol use in youth living in rural and semi-urban communities from Jalisco, Mexico, showed that youth with migration intentions had significantly higher odds of having used alcohol. This study also reported that adolescents who spoke English as a second language and had intentions to migrate had significantly higher odds of using alcohol [21].

In adolescents, immigration and nativity factors could affect integration experiences at different stages of migration, which then may have an impact on adolescents’ mental health. In the US, transnational children face numerous challenges that could expose them to different risk factors. Thus, upon arrival to the US, they are exposed to a culture and values that are more permissive than in Mexico about the use of certain drugs [22,23]. For instance, as reported by a national survey in the US, 28% of 8th graders perceive marijuana as “very easy” or “fairly easy” to obtain [24]. Recreational and medical use of marijuana is legal among adults in 18 states in the US (along with the District of Columbia), and its use is legal for medical reasons in a similar number of states [25]. Additionally, despite legally restricted access to alcohol and tobacco for US adolescents, data from a national survey showed that 45% of 8th graders think that alcohol is “very easy” or “fairly easy” to obtain, and 38.1% report fairly or very easy access to tobacco [24].

While attending school in the US, transnational children not only face sustained exposure to permissive drug use norms and opportunities but also face other challenges that might increase vulnerability to substance use [26]. Ethnic discrimination and acculturation stress have been associated with different substance abuse outcomes [26,27,28]. Among 5th grade students of Mexican origin, those living in the US for five or fewer years and who are Spanish-dominant in language spoken report more perceived ethnic discrimination, which, in turn, is associated with more recent substance use, stronger intentions to use substances, and more positive substance use expectations [28]. Additionally, transnational families, particularly those with irregular migratory status, experience chronic fear and distress that could increase their vulnerability to substance use due to the imposition of anti-immigration laws and policies at the US federal and/or state levels. These laws are a pervasive source of social stress and isolation with negative consequences for children’s development and overall well-being [29,30,31].

However, substance use vulnerability is not limited to first-generation Latinos. Data from the National Longitudinal Study of Adolescent Health show that in the US, second and third-generation adolescent Latinos report higher rates of alcohol use than their first-generation counterparts [32].

While governmental authorities in Mexico face myriad challenges in meeting the educational needs of transnational children [1,2,33], data about health-related issues in this population are limited. Thus, this study analyzed the relationship between transnational experiences in the US and the use of alcohol, tobacco, and marijuana among 7th grade students living in Mexico. We report data from a student survey in middle schools in a Mexican city bordering the US. We hypothesized that the use of these substances would be associated with students having school experience in the US and also with factors related to family dynamics, school stressors, and violence in Mexican border communities.

## 2. Material and Methods

### 2.1. Sample

The analysis utilizes data from a survey conducted in 2017 among 7th grade students in the city of Nogales, in the Mexican state of Sonora. The survey was part of a study testing the culturally adapted youth substance use prevention program *Mantente REAL*, the Spanish language version of *Keepin’ it REAL* (KiREAL) [34]. The study is a collaboration between El Colegio de la Frontera Norte (COLEF) and Arizona State University (ASU). For this study, permission was obtained from authorities, parents, and students. The study protocol was approved by the ethics committee of COLEF.

A self-administered pretest questionnaire (*n* = 1418) completed by students before the introduction of the prevention program provided data for the current study. This research activity took place during class time and a team of Mexican surveyors oversaw data collection. The questionnaire included information about substance use, students’ school experiences, and family and community issues, including the risk of violence while living in the border city. Parents provided informed consent, and students assented to be part of data collection for the study. Four public middle schools were selected, and all students enrolled in 7th grade were eligible to participate in the study (*n* = 1418). All selected schools were part of the same administrative educational system (federal). The socioeconomic status of the neighborhoods that surround the schools was relatively homogeneous (middle to low income). We analyzed three outcomes, and each variable was analyzed independently, so the number of valid cases per outcome is as follows: alcohol (*n* = 1417), tobacco (*n* = 1416), and marijuana (*n* = 1416).

### 2.2. Measures

#### 2.2.1. Dependent Variable

Students’ health behaviors were assessed with three substance use outcomes: the last 30-day frequency of use of alcohol, tobacco, and marijuana. These measures have been used previously with Mexican adolescents and were shown to be reliable [35]. Alcohol use was assessed by the question, “How many times in the last 30 days have you drunk more than a sip of alcohol (beer, wine, pulque, tequila, whisky, rum, etc.)?” Tobacco use was assessed by asking, “How many times in the last 30 days have you smoked cigarettes or tobacco?” Marijuana use was assessed by asking, “How many times in the last 30 days have you smoked marijuana (pot, weed, etc.)?”. Responses to these questions included (0) none, (1) once, (2) 2–3 times, (3) 4–9 times, (4) 10–19, (5) 20–39, and (6) 40 or more times. However, since the distributions indicated that most students were non-users or very infrequent users, we dichotomized all the outcomes into current user (1) and non-user (0).

#### 2.2.2. Independent Variables

Birthplace. To distinguish diverse groups of migrants (internal and international), we used information about birthplace to classify students into four categories: Mexican internal migrants, who include students who were born in a municipality or in a Mexican state different than Nogales, Mexico; US natives, students who were born in the US; natives of other countries, students who were born in a country other than Mexico or the US; and local natives, students born in Nogales, Mexico. Students born in the US might or might not have resided in that country for an extended amount of time.

Exposure to the US. The rationale for this variable is that having had a school experience in the US might expose the student to a permissive substance use environment, particularly US culture and values about illegal substance use. This variable includes 4 categories: (a) never in the US; (b) attended school in the US; (c) born in the US but never attended school in that country; and (d) traveled or visited the US (but never in school in the US and not born in the US).

Engagement in risky behaviors in the US. This dichotomous variable describes whether students did or did not engage in at least one of the following behaviors while living in or visiting the US: (a) used alcohol; (b) used tobacco; (c) used marijuana; (d) used other drugs; and/or (e) engaged in sexual intercourse.

Length of residence in Nogales. There are three categories: (a) those who lived in Nogales, Mexico all their life; (b) those who lived there more than 5 years; and (c) those who lived 5 years or less in Nogales.

#### 2.2.3. Social Support Variables

Social support variables include measures of household composition, the family group size in Mexico [36] and sense of belonging to school.

Household size (number of members in the household). While the assumption behind this variable is that large households increase the availability of people who would provide supervision and support to children and adolescents, a preliminary analysis of qualitative data from this study suggested that the presence in the household of certain male adults (i.e., uncles) might increase adolescents’ access to addictive substances [12].

Parents currently are living in the US. This is dichotomous variable indicates whether the student does or does not have one or both parents currently living in the US.

Financial Strain. This proxy for family socioeconomic status assesses how frequently there is not enough money in the students’ household for (a) food; (b) gasoline or transportation; (c) utilities (electricity, water, etc.); (d) school supplies; and (e) clothing that is needed. These items were measured separately on a four-point Likert scale ((0) never; (1) sometimes; (2) almost always; (3) always) and then combined into a mean scale (Cronbach’s alpha = 0.92).

Connectedness to school. This dichotomous variable assesses whether students feel a strong (enough/very much) or weak (a little/not at all) sense of belonging in their school. School connectedness can reduce the likelihood of emotional distress and of engaging in substance use, sexual intercourse, violence, and injury while drinking and driving [37].

#### 2.2.4. Urban Violence and Access to Substances

Witnessing violence around the home or neighborhood. A five-point Likert scale ((0) never; (1) once; (2) twice; (3) three times; (4) 4 or more times) was used to assess how often during their lifetime students witnessed or heard around their home or neighborhood the following: (a) shootings; (b) arrests; (c) people selling drugs; (d) someone being beaten; (e) someone stabbed; (f) someone shooting a gun; (g) gang activity; (h) someone threatened with a gun; (i) someone threatened with a knife; and (j) thefts in a house or business. These items were combined into a mean scale (Cronbach’s alpha = 0.88).

Having been victimized in the home, neighborhood or school. A five-point Likert scale ((0) never; (1) once; (2) twice; (3) three times; (4) 4 or more times) was used to assess how many times in the students’ home, neighborhood or school they have been: (a) hit; (b) kicked; (c) pushed; (d) hurt; (e) threatened with a knife or other sharp object; (f) attacked with a knife or other sharp object; (g) threatened with a gun; (h) injured by a gun; and (i) abused verbally or emotionally. These items were combined into a mean scale (Cronbach’s alpha = 0.88).

Perceived accessibility of addictive substances. Students reported how easy or difficult it would be to obtain alcohol, cigarettes, marijuana, and other illegal drugs, using a four-point Likert scale: (1) very difficult; (2) difficult; (3) easy; (4) very easy. These items were combined into a mean scale (Cronbach’s alpha = 0.87).

Sociodemographic variables. Other variables included in the analysis were sex (male = 1; female = 0), and age, measured continuously in years.

### 2.3. Statistical Analysis

For statistical analysis, we used SPSS 24 (SPSS Inc.). Descriptive statistics are presented on the 30-day prevalence of alcohol (*n* = 1417), tobacco (*n* = 1416) and marijuana use among participants. Multiple logistic regression analyses assessed the relationship between transnational experiences and consumption of addictive substances, controlling for sociodemographic variables and exposure to violence. Unadjusted and adjusted odds ratios and 95% confidence intervals (95% CI) are presented. The multiple logistic regression model generating the adjusted odds ratios included variables that, according to the literature, are associated with the consumption of alcohol, tobacco and marijuana, as well as those that in a bivariate analysis were significant predictors at a *p*-value of 0.10 or less.

## 3. Results

### 3.1. Descriptive Results

Table 1 presents the distribution of substance use outcomes, predictors, and demographic control variables. Overall, 21.7% of participants reported having used alcohol in the last 30 days, while 8.3% used tobacco and 2.4% used marijuana.

Most students were native-born in Nogales, Mexico (69.6%), 17.8% were born elsewhere in Mexico, 10.5% were born in the US, and 2.1% in a country other than Mexico or the US. While most students lived their whole lives in Nogales (62.0%), nearly one in every six (15.6%) lived there for 5 years or less. Most students had been in the US at some time (53.3%), but only 7.5% attended school in that country, and another 7.3% were born in the US but did not attend school there. Only 3.6% reported engaging in a health-related risk behavior while living in or visiting the US.

The sample had a slight majority of males (50.8%), and the average age of students was 11.9 years (SD = 0.49). Data showed that 9.5% of students had at least one parent currently living in the US, the average current household size was 3.9 members, and the average score on the financial strain measure (0.63) indicated that the families faced financial issues between “never” and “sometimes”. Almost 30% of students reported feeling little or no connection with their school.

The average perception of access to substances in their neighborhood (1.72) corresponded to responses between “very difficult” and “difficult”. The average for witnessing violent events (0.43) and having been victimized by violence (0.94) corresponded to responses between “never” and closer to “once”.

### 3.2. Relationship between Transnational Experiences and Substance Use

Table 2 shows the prevalence of recent alcohol, tobacco, and marijuana use by transnational experiences, with chi-square tests of whether the use of each substance varied by type of exposure to the US and by whether the student engaged in risky behaviors in the US. Students who attended school in the US had a significantly higher prevalence of recent use of tobacco (13.3%) and marijuana (9.5%) compared to those never in the US or never in school there. Although the differences were not statistically significant, alcohol use was also most prevalent among students who had attended school in the US. Similarly, the highest last 30-day prevalence of use of alcohol (60%), tobacco (24.6%), and marijuana (13.8%) was among students who reported engaging in health-related risk behaviors in the US.

Table 3 presents unadjusted (bivariate; [UOR]) and then adjusted (multivariate; [AOR]) odds ratios (95% CI) for the association between any 30-day use of alcohol, tobacco, and marijuana and migratory experiences and other predictors. Overall, the unadjusted analysis showed larger odds of recently using all three substances among students with transnational experiences. For instance, having attended school in the US increased the odds of recent substance use in México, compared to those students who reported only having traveled to or visited the US, although this association was statistically significant only for tobacco (UOR = 2.47, 95% CI 1.27–4.82) and marijuana use (UOR = 7.09, 95% CI 2.72–18.42). Similarly, participants who reported having engaged in health-related risk behavior in the US had significantly higher unadjusted odds of alcohol (UOR = 4.97, 95% CI 2.80–8.83), tobacco (UOR = 3.41, 95% CI 1.69–6.86), and marijuana (UOR = 8.98, 95% CI 3.67–21.99) use, compared to those that did not engage in health-related risk behaviors in the US.

Additionally, having one or both parents in the US increased the odds of recent use of marijuana (UOR = 2.54, 95% CI 1.08–5.94) among students. At the household level, a significant association was observed between family financial strain and higher odds of recent use of all three types of substances. An important source of social support to students is the level of connectedness to school: the unadjusted analysis showed that feeling little or no connection with their school increased the odds of all types of substance use (odds 1.85 to 2.24).

Similarly, the unadjusted analysis showed a significant increase in the risk of recent alcohol, tobacco, and marijuana among students reporting easier access to addictive substances in the community. Additionally, those reporting being victimized at home, school and/or in the community and having witnessed violence in the community were significantly more likely to report recent substance use. For all three substances, use was more likely among males than females and among older students.

After adjusting for all other predictors, the models with adjusted odds ratios indicated that attending school in the US was no longer associated with the use of tobacco and marijuana. However, students who had engaged in health-related risk behaviors in the US had significantly higher odds of currently using alcohol (AOR = 4.57, 95% CI 2.40–8.72) and marijuana (AOR = 3.66, 95% CI 1.19–11.24) after adjusting for other predictors.

Additionally, the adjusted models showed students feeling little or no connection with school had increased odds of alcohol (AOR = 1.52 95% CI 1.11–2.08) and tobacco (AOR = 1.71, 95% CI 1.08–2.71) use. After controlling for other variables, the odds of recent tobacco and marijuana use were significantly higher for older students.

## 4. Discussion

This study analyzed the relationship between transnational experiences in the US and the use of alcohol, tobacco, and marijuana in middle school students in a Mexican border city. We hypothesized that while living and attending school in the US, transnational adolescents were exposed to a new normative environment about risky behaviors as well as acculturative and discriminatory experiences that might influence their own patterns of substance use after relocating or returning to Mexico.

We used a cross-national framework for research on immigrant health to assess the risk-accumulation model across countries [16]. Based on this framework, we assumed that there is an interplay between receiving and returning countries that might expose transnational children to health-related risk factors that can shape patterns of initiation, current use, and cessation across migration stages [17,18,19]. In this sample, about 15% of students had the transnational migratory experience of being born in the US and/or attending school there and subsequently residing in Mexico for middle school. In addition, because Nogales is a city located directly on the US border, a large proportion of the students (38.5%) had traveled to or visited the US without the transnational experience of residing there.

Overall, the findings suggest a relationship between being a transnational adolescent and increased odds of reporting recent use of alcohol, tobacco, and marijuana in Nogales, Mexico. Transnational students who attended school in the US had the largest 30-day prevalence of use of alcohol (28.3%), tobacco (13.3%), and marijuana (9.5%) in the overall study sample. Similarly, we found a significant association between engaging in health-related risk behaviors in the US and increased odds of using alcohol, tobacco, and marijuana. Multivariate models indicated that engaging in health-related risk behaviors while in the US, rather than attending school there, was the more powerful predictor of current substance use while attending middle school in Mexico.

Studies conducted in the US have shown that while living there, transnational adolescents face numerous challenges that could increase their vulnerability to substance use. For instance, being exposed to cultural values that are more permissive about the use of certain drugs [22,23,24], along with facing discrimination and pervasive anti-immigrant laws [28,29,31]. Although this study did not assess or control for these potential explanatory factors, most of these influences have been related to substance use among Mexican origin or other Latino children in the US [23,28].

Additionally, the deportation of parents to Mexico puts adolescents—regardless of citizenship—and their families at risk of economic instability, school disruption, and negative health outcomes [1,5]. As reported above, since 2008, nearly half a million children and adolescents have been deported to Mexico. Most of these children were born in the US, and currently confront diverse integration barriers in Mexican schools and communities [1,2].

Although there was a bivariate relationship between having a parent or parents living in the US and the child’s use of marijuana in Mexico, in the adjusted analysis, this relationship was no longer statistically significant. There is evidence that parental absence can increase the risk of adolescent use of illegal drugs such as marijuana but not alcohol use [34]. Research about the effect of parental migration on children left behind is inconclusive. For instance, while a meta-analysis of the effect of parental migration on the health of children and adolescents left behind found an increase in the risk of several mental health issues such as depression, suicidal ideation, anxiety, and substance use [38], studies on the impact of remittances sent by migrating parents suggest they have an important role in promoting children’s health [39,40].

Though generalized poverty at the community level and household financial strain are risk factors for substance use among children and adolescents, the link between these variables is complex. Some evidence suggests that affluent adolescents may be more prone to experiment with and have access to substances, such as marijuana [41,42,43]. In this study, after adjusting for other predictors, household financial strain was related to an increased probability of use of all the substances analyzed. This finding is consistent with a study that used data from a Mexican national survey, which found that 5th and 6th grade students had a lower risk for tobacco use and intentions to use substances when reporting a subjective sense of occupying higher socioeconomic status [44]. Additionally, household economic strain, particularly having experienced poverty during childhood, was linked to regular tobacco use in adulthood [45]. Similarly, a study conducted among students in 28 European countries found a strong direct relationship between some patterns of cannabis, cocaine, and heroin use and family characteristics such as lower socioeconomic status and lower parental education [41].

School connectedness and school failure are important predictors of substance use and abuse among adolescents [46,47]. We reported that not feeling connected to school was significantly linked with alcohol and tobacco use but not marijuana use. A combination of factors might explain the lack of association with marijuana use. In this study, recent marijuana use was reported by only 2.4% of students and 13.3% of those who attended school in the US. Moving abruptly to a Mexican northern border city exposes transnational children to new environments that might increase their vulnerability to substance use. A study conducted with transnational children in Tijuana showed that after arriving in Mexico and attending school, transnational students disliked their schools because they lacked policies that supported them academically, and their educational infrastructure was of poor quality [1,40].

In our study, the perception of easy access to addictive substances in the community and having witnessed violent events around home or school were also associated with the use of alcohol, tobacco, and marijuana. Our measures of exposure to violence do not distinguish whether it occurred in the US or Mexico. However, northern Mexican border states are strategic in the trafficking of drugs into the US, and this region has experienced an increase in drug trafficking-related violence in recent decades, with consequences for the quality of life and well-being of border communities [15]. For this reason, Valdez et al. [48] characterize border communities as a “toxic” environment because drug trafficking seems “normal” and provides access to illegal drugs in the region. Criminal insecurity and violence are other intense challenges facing border communities [9,10,15,49]. In a 2022 national survey about the perception of insecurity, more than half of adults (51.7%) described Nogales as an insecure place [49].

This study contributed to knowledge of substance use-related risk factors among transnational adolescents living in Mexico. We found an interplay between the receiving and returning countries that exposed transnational adolescents to an accumulation of risk factors in both settings. Still, our findings have limited generalizability due to the study being conducted in a small number of schools, the relatively homogenous socioeconomic status of the students, and the focus only on 7th grade students in early adolescence. Another limitation was that students’ addictive substance use and risky behaviors in the US were assessed based on self-report and not independently confirmed. Nevertheless, the study had a high response rate, used validated questionnaire items, and produced findings consistent with expected relationships.

This study fills an important gap in understanding the relationship between transnational experiences in the US and the use of alcohol, tobacco, and marijuana in students attending school in Mexico. Although other studies have shown an association between substance use across different international migration stages in adults, to the best of our knowledge, this study is one of the first to assess the accumulation of health-related risk factors among young transnational students. Findings from this study illustrate the importance of using a cross-national framework to assess the accumulative effects of risk factors on immigrant health.

## 5. Conclusions

Although additional studies are needed to better understand the accumulation of risk factors in different stages of migration as well as how each migration stage is linked to patterns of substance use, there are evidence-based prevention interventions that could be applied to reduce tobacco, marijuana, and alcohol use among vulnerable and understudied youth in a transnational context [50,51]. However, we need to evaluate the degree to which these interventions will achieve the same outcomes as with other populations without some level of adaptation [52]. Future research in this area would benefit from a focus on dissemination and implementation science that can guide and test adaptations based on core components of evidence-based interventions in the contexts where transnational children arrive [53]. In addition, prevention with this population can be advanced by focusing on the contextual factors that could be addressed through strategies supporting the adoption, implementation, and sustainability of policy, practice, and curricular-based initiatives [54]. Specifically, school authorities and local governments could apply these findings to inform their educational policies addressing the unique needs of transnational students. At the school level, teachers and other school personnel could benefit from strategies focusing on capacity building to work with a growing diversity of students’ psychosocial needs in their classrooms [54]. The findings also provide leads for future research studies and opportunities to advance prevention science on the Mexico–US border that could be applicable to transnational populations in other countries. There is a role for prevention and implementation science to better inform interventions tailored to the needs of young people searching for their own identity and a sense of home. Investing in these research areas can have important implications for the well-being of students and society at large.

## Figures and Tables

**Table 1 ijerph-21-00043-t001:** Distributions for substance use outcomes, predictors, and demographic controls (*n* = 1418).

Variables	*n* (%)	Mean (SD)
**Outcomes variables**		
30-day alcohol use (*n* = 1417)		
No	1110 (78.3)	
Yes	307 (21.7)	
30-day tobacco use (*n* = 1416)		
No	1298 (91.7)	
Yes	118 (8.3)	
30-day marijuana use (*n* = 1416)		
No	1382 (97.6)	
Yes	34 (2.4)	
**Migration experience/exposure to the US**		
Birthplace		
Nogales	983 (69.6)	
Another municipality in Mexico	252 (17.8)	
US natives	149 (10.5)	
Other country	29 (2.1)	
Exposure to the US		
Never in the US	662 (46.7)	
School in the US	106 (7.5)	
Born in the US but never attended school in the US	103 (7.3)	
Traveled to or visited the US (not schooled or born in the US)	547 (38.6)	
Engaged in risky behaviors in the US		
Yes	50 (3.6)	
No	1351 (96.4)	
Length of residence in Nogales		
≤5 years	219 (15.6)	
+5 years	315 (22.4)	
Native/entire life	873 (62.0)	
**Social support**		
Parent(s) currently living in the US.		
Yes	135 (9.5)	
No	1283 (90.5)	
Household size		3.93 (0.91)
Financial strain		0.63 (0.83)
Connectedness to school		
Enough/very much	904 (70.2)	
A little/no at all	383 (29.8)	
**Urban violence and perception access to substance**		
Perceived accessibility of addictive substances		1.72 (0.85)
Having been victimized in the home, neighborhood, or school		0.94 (0.89)
Witnessing violence around the home or neighborhood		0.43 (0.60)
**Demographics**		
Age		11.93 (0.49)
Sex		
Female	692 (49.2)	
Male	715 (50.8)	

**Table 2 ijerph-21-00043-t002:** Prevalence of 30-days substance use by students’ transnational migratory experiences.

	Alcohol	Tobacco	Marijuana
Exposure to the US			
Never in the US	144 (21.8)	62 (9.4) *	13 (2.0) ***
School in the US	30 (28.3)	14 (13.3)	10 (9.5)
Born in the US but never attended school in US	19 (18.4)	10 (9.7)	3 (2.9)
Traveled to or visited US (not schooled or born in US)	114 (20.8)	32 (5.9)	8 (1.5)
Engaged in risky behaviors in the US			
No	264 (19.8) ***	98 (7.3) ***	22 (1.6) ***
Yes	39 (60.0)	16 (24.6)	9 (13.8)

*** *p* < 0.001, ** *p* < 0.01, * *p* < 0.05.

**Table 3 ijerph-21-00043-t003:** Unadjusted and adjusted odds ratios (95% confidence intervals) for the association between recent substance use and migration experiences, social support, access to substances, urban violence, and demographics.

Variables	Substance Use in the Last 30 Days					
	Alcohol		Tobacco		Marijuana	
	UOR (95% CI)	AOR (95% CI)	UOR (IC 95%)	AOR (95% CI)	UOR (95% CI)	AOR (95% CI)
**Migration experience/exposure to the US**						
Exposition to the USA		n/a		n/a		n/a
Never in the US	1.05 (0.80, 1.39)		1.66 (1.07, 2.59) *		1.35 (0.55, 3.28)	
School in the US	1.49 (0.93, 2.40)		2.47 (1.27, 4.82) **		7.09 (2.72, 18.42) ***	
Born in the US but never attended school in US	0.85 (0.50, 1.47)		1.73 (0.82, 3.64)		2.02 (0.52, 7.75)	
Traveled to or visited US (not schooled or born in US)	1.0		1.0		1.0	
Engaged in risky behavior in the US				n/a		
Yes	4.97 (2.80, 8.83) ***	4.57 (2.40, 8.72) ***	3.41 (1.69, 6.86) ***		8.98 (3.67, 21.99) ***	3.66 (1.19, 11.24) *
No	1.0	1.0	1.0		1.0	
Length of residence in Nogales		n/a		n/a		n/a
≤5 years	1.12 (0.78, 1.60)		1.17 (0.70, 1.96)		1.82 (0.81, 4.06)	
+5 years	1.13 (0.82, 1.54)		0.92 (0.56, 1.48)		0.55 (0.18, 1.62)	
Native/Whole life	1.0		1.0		1.0	
**Social support**						
Parents currently living in the US		n/a		n/a		n/a
Yes	1.13 (0.74, 1.72)		1.43 (0.80, 2.53)		2.54 (1.08, 5.94) *	
No	1.0		1.0		1.0	
Household size	0.97 (0.84, 1.11)	n/a	0.94 (0.76, 1.16)	n/a	0.83 (0.57, 1.22)	
Financial strain	1.34 (1.15, 1.56) ***	1.30 (1.09, 1.54) **	1.36 (1.09, 1.70) **	1.36 (1.07, 1.74) *	1.63 (1.14, 2.34) **	1.59 (1.04, 2.44) *
Connectedness to school						n/a
Enough/very much	1.0	1.0	1.0	1.0	1.0	
A little/no at all	1.85 (1.40, 2.44) ***	1.52 (1.11, 2.08) **	1.99 (1.33, 2.99) **	1.71 (1.08, 2.71) *	2.24 (1.07, 4.69) *	
Perceived accessibility of addictive substances	1.43 (1.24, 1.65) ***	1.30 (1.11, 1.54) **	1.84 (1.52, 2.24) ***	1.74 (1.39, 2.18) ***	2.61 (1.87, 3.65) ***	2.22(1.49, 3.31) ***
Having been victimized in the home, neighborhood, or school	1.57 (1.29, 1.92) ***	n/a	1.66 (1.28, 2.15) ***	n/a	1.81 (1.21, 2.72) **	
Witnessing violence around the home or neighborhood	1.68 (1.47, 1.92) ***	1.62 (1.39, 1.89) ***	1.79 (1.50, 2.14) ***	1.70 (1.38, 2.10) ***	1.85 (1.37, 2.49) ***	1.68(1.16, 2.43) **
**Demographics**						
Age	1.65 (1.28, 2.12) ***	n/a	2.24 (1.59, 3.16) ***	1.89 (1.26, 2.83) **	2.65 (1.57, 4.45) ***	2.32 (1.23, 4.37) **
Sex		n/a		n/a		n/a
Female	1.0		1.0		1.0	
Male	1.53 (1.18, 1.98) ***		1.51 (1.03, 2.29) *		1.80 (0.88, 3.67)	

*** *p* < 0.001, ** *p* < 0.01, * *p* < 0.05.

## Data Availability

Although not publicly available, data can be made available on request subject to restrictions to ensure confidentiality. The data presented in this study are available on request from the corresponding author. Given that the study is a collaboration between El Colegio de la Frontera Norte (COLEF) and Arizona State University (ASU), quantitative and qualitative data are stored securely at both universities.

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
