# Peer review of "Urban Violence, Migration and Alcohol, Tobacco, and Marijuana Use among Transnational Students in Northern Mexico"

_ijerph, 2023, doi:10.3390/ijerph21010043_

Round 1

Reviewer 1 Report

Comments and Suggestions for Authors

As a whole, the article presented appears to be of excellent quality. Its introduction is well achieved and supported by previous literature, the methodology was well presented and the results well discussed. However, like any work, it is important to mention some points that can be improved.

1.       Line 49: “children’s mental health and well-being” instead of “children’s mental health and the well-being”;

2.       Although the information provided in the introduction is extremely relevant, it would be important to explore in greater depth the risk factors associated with young Mexicans attending school in Mexico, similar to what was done with young Mexicans attending school in the US;

3.       Much of the information in the "Participants" subsection should be in a section titled "Procedures";

4.       Line 151: there is a “?” missing before the closing quotation mark;

5.       It was not clear what the cut-off point was at which a participant was considered a user for any type of substance;

6.       Lina 192: after “always”, the coding number is missing;

7.       Line 242: an equal sign after “SD” is missing;

8.       Table 3 must be reformatted in order to comply with the standards;

9.       Line 333: This sentence is not finished;

10.   The article must be reviewed in full, in order to correct some grammatical errors and gaffes (e.g., extra spaces).

Comments on the Quality of English Language

Some minimal errors can be corrected.

Reviewer 2 Report

Comments and Suggestions for Authors

Very interesting argument to highlight the context of violence and mental health challenges of migrant children, probably, whose parent has been deported.

Find below a few observations

Method

How many copies of pretest questionnaire was administered and where and whom was it administered to ---line 130

Whose parent provided consent (is it those that were deported?) line 134

How many federal schools do we have in the country and how many was selected for the study  ....line 136

independent variable: Consider adding age of respondent

What effort was put in place to reduce bias

What is the policy implications of the study

State the limitation and strength of the study

Reviewer 3 Report

Comments and Suggestions for Authors

Well written article

Suggest the following minor edits mostly for contextualizing the study for readers:

(1) At the beginning of the narrative under the subheading The risk-accumulation model a brief explanation of the cross-national framework.  

(2) Was the 2017 survey just focused on Nogales or was the Nogales data part of a larger country wide border town survey - please clarify.  If the data were part of a larger country-wide survey then why Nogales and not other border towns?

Again, minor suggestions is all.  Great article!

Round 2

Reviewer 2 Report

Comments and Suggestions for Authors

The variables under consideration is interesting. I observe efforts to improve the manuscript. However, the English language is poor.  I advise you work on context editing.

Good luck

Comments on the Quality of English Language

The English language is poor. 

Author Response

Two native English speakers proofread the manuscripts and some editing was made to ensure and improve the manuscript’s readability.

Reviewer 3 Report

Comments and Suggestions for Authors

Revision is fine.

Author Response

We are attaching the revised manuscript “Urban Violence, Migration and Alcohol, Tobacco and Marijuana Use among Transnational Students in Northern Mexico” addressing all the observations and recommendations by reviewers and the editor.